# Public Health Risk Associated with Botulism as Foodborne Zoonoses

**DOI:** 10.3390/toxins12010017

**Published:** 2019-12-30

**Authors:** Christine Rasetti-Escargueil, Emmanuel Lemichez, Michel R. Popoff

**Affiliations:** Institut Pasteur, Département de Microbiologie, Unité des Toxines Bactériennes, CNRS ERL6002, 75724 Paris, France; christine.rasetti-escargueil@pasteur.fr (C.R.-E.); emmanuel.lemichez@pasteur.fr (E.L.)

**Keywords:** botulism, *Clostridium botulinum*, *Clostridium baratii*, *Clostridium butyricum*, botulinum neurotoxin

## Abstract

Botulism is a rare but severe neurological disease in man and animals that is caused by botulinum neurotoxins (BoNTs) produced by *Clostridium botulinum* and atypical strains from other *Clostridium* and non-*Clostridium* species. BoNTs are divided into more than seven toxinotypes based on neutralization with specific corresponding antisera, and each toxinotype is subdivided into subtypes according to amino acid sequence variations. Animal species show variable sensitivity to the different BoNT toxinotypes. Thereby, naturally acquired animal botulism is mainly due to BoNT/C, D and the mosaic variants CD and DC, BoNT/CD being more prevalent in birds and BoNT/DC in cattle, whereas human botulism is more frequently in the types A, B and E, and to a lower extent, F. Botulism is not a contagious disease, since there is no direct transmission from diseased animals or man to a healthy subject. Botulism occurs via the environment, notably from food contaminated with *C. botulinum* spores and preserved in conditions favorable for *C. botulinum* growth and toxin production. The high prevalence of botulism types C, D and variants DC and CD in farmed and wild birds, and to a lower extent in cattle, raises the risk of transmission to human beings. However, human botulism is much rarer than animal botulism, and botulism types C and D are exceptional in humans. Only 15 cases or suspected cases of botulism type C and one outbreak of botulism type D have been reported in humans to date. In contrast, animal healthy carriers of *C. botulinum* group II, such as *C. botulinum* type E in fish of the northern hemisphere, and *C. botulinum* B4 in pigs, represent a more prevalent risk of botulism transmission to human subjects. Less common botulism types in animals but at risk of transmission to humans, can sporadically be observed, such as botulism type E in farmed chickens in France (1998–2002), botulism type B in cattle in The Netherlands (1977–1979), botulism types A and B in horses, or botulism type A in dairy cows (Egypt, 1976). In most cases, human and animal botulisms have distinct origins, and cross transmissions between animals and human beings are rather rare, accidental events. But, due to the severity of this disease, human and animal botulism requires a careful surveillance.

## 1. Introduction

Botulism is a severe disease of man and animals that is characterized by flaccid paralysis leading to respiratory distress and death in the most serious cases. This disease is due to a potent neurotoxin, botulinum neurotoxin (BoNT), that is produced by toxigenic *Clostridium botulinum* strains and more rarely by other *Clostridium* and non-*Clostridium* species. 

Accumulation of BoNT in contaminated food during the period of food preservation before consumption, in conditions allowing bacterial growth, is responsible for human food borne botulism that is the main form of botulism in most countries. In certain circumstances, *C. botulinum* can colonize the intestinal tract and produce BoNT in situ. Thus, *C. botulinum* spore ingestion in newborn or young infants up to one year old, where the microbiota is not fully developed or not fully functional, can result in *C. botulinum* growth and BoNT production in the intestinal tract, and subsequently infant botulism. Infant contamination by *C. botulinum* is mainly from the environment, notably household dust [1,2,3]. Botulism by intestinal colonization is less frequent in adults [4,5]. Conditions that result in microbiota perturbation such as abdominal surgery in one or two weeks before the ingestion of contaminated food, antibiotic treatment or malformation of the intestinal tract (Meckel’s diverticulum) are favorable to *Clostridium* growth in the intestine. Other rare forms of botulism in humans consist in wounds, inhalation and iatrogenic botulism. In contrast to humans, botulism is much more frequent in animals, raising the risk of transmission to humans.

## 2. Diversity of Botulinum Neurotoxins

BoNTs are divided into several toxinotypes according to the inhibition of their biological activity by specific neutralizing antibodies. Each toxinotype is neutralized by its corresponding antiserum and not by the sera against the other toxinotypes. Mouse bioassay is still the reference method for BoNT identification and typing. Thereby, seven toxinotypes are classically identified, being the toxinotypes A, B, C, D, E, F and G. BoNT genes have been sequenced in numerous *C. botulinum* strains, and amino acid sequence variations are observed between BoNTs of each toxinotype. Based on these amino acid sequence variations, BoNTs of each toxinotype are subdivided into subtypes. The threshold of amino acid sequence variation that defines two distinct subtypes is still matter of debate. A threshold of 2.6% was applied for the subtypes of BoNT/A. However, certain BoNT subtypes, notably from types B and E, show only 0.9–2.1% amino acid sequence difference, but they were assigned to distinct subtypes. Until now, 41 subtypes have been identified [6]. Amino acid sequence variations in BoNT subtypes might impact on their biological activities and on their recognition by monoclonal antibodies. Whereas, neutralizing polyclonal antibodies, such as the classical anti-BoNT equine sera, neutralize all the BoNT of the same type, anti-BoNT monoclonal antibodies might recognize only certain subtypes and no other from BoNTs of a same toxinotype [7,8]. 

BoNT/C and BoNT/D show only two variants which are called mosaic BoNT/CD and BoNT/DC. These variants mostly result from hybrids between BoNT/C and BoNT/D. BoNT/CD contains two thirds of BoNT/C at the N-terminal, and one-third at the C-terminal of BoNT/D, whereas BoNT/DC is composed of two thirds of BoNT/D at the N-terminal and the C-terminal has one-third of BoNT/C [9]. The antisera against BoNT/C and BoNT/D, respectively, neutralize specifically the corresponding BoNT/C or BoNT/D. However, BoNT/CD is neutralized by both sera anti-BoNT/C and -BoNT/D [10,11], whereas BoNT/DC is neutralized by anti-BoNT/D and only partially by anti-BoNT/C [12].

Novel BoNT types have been identified in the recent years. BoNT/H (F/A or H/A) was claimed not to be neutralized by any of the already known anti-BoNT sera, justifying its assignment to a novel type. However, BoNT/H is neutralized by the BoNT/A antiserum, but with a higher dose of antiserum than that required for the neutralization of BoNT/A [13,14,15]. More recently, a novel BoNT called BoNT/X was characterized in the *C. botulinum* strain that also produces a BoNT/B2. BoNT/X shares a low level of amino acid sequence identity with the other BoNT sequences, and it is not recognized by the antibodies against the other BoNT types [16].

Related BoNT sequences have also been identified in non-*Clostridium* species such as BoNT/Wo or BoNT/I in the Gram-negative *Weisella oryzae* from fermented rice, BoNT/J and BoNT/Cp1 in the Gram-negative *Chryseobacterium piperi* from sediment, or BoNT/En or eBoNT/J from the Gram-positive *Enterococcus faecalis* isolated from a cow [17,18,19,20]. However, the production of these BoNT-related proteins by these non-*Clostridium* species, as well as their in vivo toxicity, have not yet been confirmed.

## 3. Botulinum Neurotoxin Producing Bacteria

Until now, biologically active BoNTs are produced by Gram-positive, anaerobic and sporulating bacteria from the genus *Clostridium.* Clostridia producing a neurotoxin responsible for flaccid paralysis were assigned to the *C. botulinum* species. However, *C. botulinum* is a heterogeneous bacterial species which is subdivided into four groups. Most of the *C. botulinum* strains produce only one type of BoNT, whereas certain rare strains produce two BoNT types (bivalent strains) or three BoNT types (trivalent strains). According to the BoNT type produced, the four groups of *C. botulinum* are:group I *C. botulinum* types A, H (F/A or H/A) and proteolytic *C. botulinum* B and F strainsgroup II *C. botulinum* E and glucidolytic, non-proteolytic *C. botulinum* B, and F strainsgroup III *C. botulinum* C, D, CD and DCgroup IV C. argentinense or C. *botulinum* G

Other atypical strains from other *Clostridium* species are neurotoxigenic, and have been assigned to:groups V *C. baratii* strains producing BoNT/F7groups VI *C. butyricum* strains producing BoNT/E4 and E5

On a bacteriological level, *C. botulinum* strains from groups I to IV correspond to distinct bacterial species, based on their phenotypic and genetic differences. Whole genome sequencing and phylogenetic analysis confirm the distinction of different *Clostridium* species producing BoNTs, and the name of *C. parabotulinum* has been proposed for the strains of group I, of *C. botulinum* for the strains of group II, *C. novyi* sensu lato for the strains of group III, and *C. argentinense* for group IV [21]. 

The particularity of group III *C. botulinum* strains is that the neurotoxin genes, *bontC* and *bontD*, are localized on phages which are not integrated into the chromosome [22,23,24,25]. Thus *bont*-harboring phage can be easily lost during *C. botulinum* cultures which become no longer toxic. Very often, the group III *C. botulinum* colonies isolated from biological or environmental samples are non-toxinogenic derivative strains without the *bontC* or *bontD* harboring phage.

It is noteworthy that the strains of each group are related to non-neurotoxigenic *Clostridium* species. Notably, the *C. botulinum* of group III is related to *Clostridium novyi* and *Clostridium haemolyticum*. *C. botulinum* C and D phages containing the neurotoxin genes (*bontC* and *bontD*, respectively) as well as plasmids, can be interchanged between group III *C. botulinum, C. novyi* and *C. haemolyticum* strains [26,27,28,29,30,31,32,33,34].

## 4. Animal Botulism

### 4.1. Group III C. botulinum and Animal Botulism

BoNT/A to G can attack all the vertebrates, but the invertebrates are resistant due to lack of specific receptors on their neuronal cell surface [35]. Thereby, human and all vertebrates are susceptible to develop botulism A to G, but the sensitivity of each animal species is variable according to the BoNT type (Table 1). The host sensitivity depends of the presence/abundance of functional BoNT receptors on neuronal cell surfaces, and also of the presence or absence of cleavable intracellular target isoforms (SNAP25, VAMP). The sensitivity of the main domestic animal species to BoNTs according to experimental tests is summarized in Table 1. Recently, a novel BoNT called paraclostridial mosquitocidal protein1 (PMP1) produced by a *Paraclostridium bifermentans subsp Malaysia* strain has been found to be specific of *Anopheles* mosquitos [36]. This is the first BoNT type that has been identified to target an invertebrate species.

Historically, strains isolated from chickens with botulism in the United States and cattle in Australia were identified as belonging to a novel type, termed type C by Bengtson and Seddon in 1922 [37,38]. A distinct microorganism was isolated from a bovine with botulism in South Africa by Meyer and Gunnison (1928) and was called type D [39]. Then, *C*. *botulinum* C and D were mostly isolated or identified in animal botulism. Later, the mosaic BoNT/DC and BoNT/CD were characterized [9]. 

Based on their biochemical properties, the strains producing BoNT/C and BoNT/D were assigned to the *C. botulinum* group III. Since the optimum temperature of growth of group III *C. botulinum* is about 40 °C, outbreaks of botulism types C and D in animals are mainly observed in the hot countries and during the hot periods in the temperate countries.

### 4.2. Group III C. botulinum are Mainly Prevalent in Birds and Cattle

Botulism is mainly prevalent in birds (wild and farmed birds), and cattle and causes important economic losses every year throughout the world. Botulism is more sporadic in the other animal species.

Albeit botulism type C and D are typically considered as animal botulism, the different animal species show different sensitivity to BoNT/C and D (Table 1). For example, birds are sensitive to BoNT/C and resistant to BoNT/D, whereas cattle are sensitive to both BoNT/C and BoNT/D. Among birds, chickens, pheasants and turkeys are more sensitive to BoNT/C than ducks (Table 1).

In the previous works prior to the genetic characterization of the mosaic *C. botulinum* C/D and D/C [9,52], only BoNT/C and BoNT/D were identified by neutralization with the standard anti-BoNT/C and anti-BoNT/D sera, but it was noted that certain BoNT samples were neutralized by both anti-BoNT/C and anti-BoNT/D sera. BoNT from *C. botulinum* C/D is neutralized by standard anti-BoNT/C and anti-BoNT/D sera, and BoNT from *C. botulinum* D/C is neutralized only by anti-BoNT/D serum [11,12].

Wild birds, mainly waterbirds such as duck, and farmed birds, are frequently affected by botulism. Indeed, frequent and important outbreaks of botulism in water birds are regularly reported in Europe, North America and Japan [11,53,54]. For example, in Spain, 13 outbreaks of botulism in wild birds caused the deaths of about 20,000 birds, including more than 50 species between 1978 and 2008 [55]. In France during the period 2000 to 2013, 129 outbreaks of botulism have been reported in wild birds and 396 in farmed birds [53]. In an investigation in 17 flocks affected with botulism, the mortality was from 2.8% to 35%, each flock containing 10,000 to 20,000 farmed birds [56]. In large outbreaks of type E botulism in the Great Lakes (US) between 2000 and 2008, the mortality was estimated to around 68,000 wild birds [57]. Botulism in birds is mostly type C or D, as tested by BoNT typing in blood or organ samples and C/D by molecular biology [10,11,55,58,59,60,61]. As an example, data from our laboratory show a prevalence of the toxinotype neutralized by both anti-BoNT/C and D in farmed birds, and a wider distribution of toxinotypes in wild birds and cattle (Table 2). More recent investigations by genotyping indicated that the mosaic type C/D is more frequently associated with botulism in birds and the mosaic D/C in cattle, in agreement with the results reported by other laboratories [11,12,56,58,59,60,62,63,64]. 

### 4.3. Transmission of Botulism between Animal Species

Botulism is not a contagious disease, that is, there is no direct transmission of botulism from a diseased animal to a healthy one. Contamination results from an ingestion of vegetative cells or spores and/or preformed BoNT in the environment, notably in food. Indeed, cadavers of animals which have died of botulism, or of healthy carriers of *C. botulinum* in their intestine, constitute an excellent environment for *C. botulinum* growth and toxin production. Therefore, they represent a major source of contamination and dissemination of animal botulism. Thus, animal botulism results either from the ingestion of preformed BoNT in food, or from intestinal colonization and subsequent toxin production in the intestine. Carrion of small animals (rodents, birds, cats) in food or decaying carcasses that are chewed by phosphorus-deficient cattle are the main source of botulism by intoxication in animals [66,67,68]. BoNT production can also occur in fermented cereals and inadequately stored silages with pH > 4.5 [69,70,71]. Botulism by intestinal colonization is a frequent form of botulism in farmed birds and cattle. Thereby, in most outbreaks of bovine or bird botulism, BoNT was not detected in silages or industrial livestock feed based on dry, crushed cereals containing *C. botulinum* spores that were responsible for botulism [72,73,74,75,76]. In addition, botulism cases are usually observed up to two weeks after the withdrawal of the contaminated food supporting a contamination by intestinal colonization. Young animals seem more susceptible to botulism by intestinal colonization as in humans. Thereby, botulism by intestinal colonization is more prevalent in foals, whereas food borne botulism by intoxication seems more frequent in adult horses [77]. Microbial changes and composition of the fecal microbiota influence the development of *C. botulinum* in the digestive tract. The presence of *C. botulinum* in the rumen of dairy cows in Germany was associated with a significant greater prevalence of some bacterial species (*Bacteroides* spp., *Clostridium histolyticum* group, Alfa-proteobacteria, Gamma-proteobacteria and sulfate-reducing bacteria) and with lower counts of *Euryaracheota* and protozoa [78].

The diseased animals eliminate with their feces *C. botulinum* vegetative cells/spores which accumulate in their own environment. The promiscuity of animals in a flock, notably in industrial farms that can contain several thousands of animals in a restricted space associated with favorable conditions of temperature and wetness, facilitate the transmission of botulism between animals via the environment. The contaminated environment of a flock with botulism, notably manure, spread on a new environment, can be responsible for additional outbreaks of animal botulism. This is the case of the transmission of botulism via litters from the industrial farms of birds with botulism spread on grazing pasture. Thereby, botulism is frequent in cattle in areas where dense populations of cattle flocks are in the proximity of industrial farms of birds. Preparations of silages from these contaminated pastures, notably if the pH is not sufficiently acidic, facilitate the multiplication of *C. botulinum* and the subsequent transmission of botulism. However, botulism in birds are mostly due to *C. botulinum* C/D, whereas cattle botulism mainly results from *C. botulinum* D/C [11,12,58,79,80,81,82]. 

Albeit chickens are resistant to BoNT/D, they can host in their digestive tract BoNT/D-producing clostridium. Thereby, they can contaminate the environment via their feces and transmit botulism to other BoNT/D-sensitive species such as cattle. This has been evidenced in the first outbreaks of cattle botulism in France in 1979–1981 [73,83], where BoNT/D and *C. botulinum* D were characterized in the digestive tract of cattle that died with symptoms of botulism. *C. botulinum* D was identified in laying hens from an industrial farm without suspicion of botulism, that was close to the cattle flocks and in the industrial food for hens/chickens, as well as in meat meal samples prepared with cadavers of quartering used for animal foods [73,83]. Type D botulism in cattle has also been associated with the spreading of broiler litter on pasture in the United Kingdom (UK), albeit no botulism was observed in the broiler farms [84]. More recently, *C. botulinum* D/C was detected in a healthy poultry farm (litter, farm dust), silage and pasture where chicken manure had been spread, as well as in cattle that died of botulism in two farms at the proximity of the poultry farm [85]. Therefore, these observations supported that an animal species resistant to one type of botulism can be a healthy carrier of the corresponding *C. botulinum*, and therefore the source of contamination of other sensitive animal species. Thus, the healthy carriers constitute a critical reservoir of certain *C. botulinum* types where they can abundantly multiply and subsequently target sensitive animal species. 

Outbreaks of type C botulism in cattle have been associated with the spreading on grazing areas of poultry litter that might contain poultry carcasses, or with dietary supplement consisting of ensiled poultry litter [86,87,88,89,90,91,92,93,94]. In addition, botulism in cattle might result of hay, haylage, or silages contaminated with the decaying carcasses of cats or rodents [66,67]. Cadavers of small animals such as rats and cats are appropriate environments for *C. botulinum* growth and toxin production. They may contaminate livestock feed and subsequently cause botulism outbreaks [66]. For example, in a large outbreak in France, 80 of 110 dairy cows died. The animals received grass silage contaminated with poultry litter [72]. In Finland, nine out of 90 cattle fed non-acidified silage contaminated by carrion died [66]. In Brazil, a massive outbreak of botulism type C caused the death of 1100 steers from a flock of 1700 which had been fed corn silage in two weeks. The origin of the corn silage contamination was not determined [70].

Transmission of botulism might also occur between wild type and farmed animals. This is mainly critical between wild and farmed birds in which botulism is frequent in both types of animals, mainly waterfowl. However, most of the bird farms are sufficiently isolated, avoiding direct contact between wild birds and farmed birds. Moreover, the occurrence of botulism in wild birds is more seasonal than in farmed birds. Wild bird botulism mainly occurs in the hot periods, summer time and autumn in the temperate countries, whereas botulism in farmed birds is observed in any season [53]. An outbreak of type C botulism in cattle in Canada has been suspected to be associated with botulism in waterfowl [95]. It seems that this mode of interspecies botulism transmission is only occasional. 

## 5. Risk of Transmission of Animal Botulism to Human

### 5.1. Human Botulism with Group III C. botulinum

In contrast to animal botulism due to the toxigenic strains of group III *C. botulinum*, which is very frequent throughout the world, types C and D botulism are very rare in humans. Only eigth outbreaks (15 cases) of type C botulism and one outbreak of type D botulism were reported in the literature (Table 3). However, type C botulism in humans was confirmed in only 4 outbreaks (8 cases). These cases of human botulism were food borne botulism, except one case of infant botulism in Japan, in which the contamination was probably due to spores from the environment, since no food contamination related to this infant botulism case was reported [96]. Among the types C and D human cases of botulism, three were severe forms. One patient in the USA died subsequently to clinical symptoms of botulism. BoNT/C and a strain of *C. botulinum* C were identified in the patient’s stomach content [97,98]. In one outbreak in France, two patients, a 69 years old woman and a 11 years old girl, developed severe symptoms of botulism including diplopia, ptosis, dysphagia, limb paralysis and respiratory distress in the woman, who deceased following infectious complications. The little girl showed diplopia, ptosis, asthenia and cardiac arrest 11 days after the onset of symptoms. She recovered after two months of hospitalization [99]. It is noteworthy that most of these human type C botulism cases were described before 1970, in a period where the typing of *C. botulinum* was not accurately defined. The distinction of Cα and Cβ types was still confusing. The outbreaks reported in Rhodesia and the USRR have not been further confirmed (Table 3).

In two outbreaks, *C. botulinum* type C was identified in home-made “paté”, without indication of the origin of the meat, pork or chicken, and the *C. botulinum* type D strain was isolated from a pork product. It is not clear whether the contamination occurred from the animal products or from the environment during the food processing (Table 3). In the case of the Guyana outbreak, botulism symptoms occurred in a man who consumed ill chickens from his farm. Botulism type C/D was identified in his poultry farm (BoNT neutralized by both types C and D antisera in the serum samples of two hens and one duck). The patient recovered rapidly in one or two days [100]. This observation raises the possibility of type C botulism transmission from chicken to human beings, but supports that humans are poorly sensitive to type C food borne botulism.

The unique outbreak of human type D botulism concerned a group of eight persons who have eaten home-made, uncooked, salted ham. Five of them developed vomiting and diarrhea, probably due to other contaminants than *C. botulinum* in the ham, and another one had blurred vision during 10 days, which is typical of the mild form of botulism. Another patient showed diplopia, dry mouth, dysphagia, constipation, asthenia, but then recovered in one month [101]. The *C. botulinum* strain isolated from the ham, called 1873, is considered as a reference type D *C. botulinum* [102,103]. The extreme rare prevalence of type D human botulism is supported by the poor sensitivity of humans to BoNT/D. Thereby, injection of BoNT/D in the Extensor Digitorum Brevis muscle of human volunteers was ineffective to induce significant muscle paralysis [104].

### 5.2. Human Botulism with Group II C. botulinum from Healthy Carrier Animals

Animals can be healthy carriers of *C. botulinum*, and either they are resistant to certain BoNT types, or they contain a low number of *C. botulinum* bacteria in their digestive tract. *C. botulinum* can possibly grow in the intestine of healthy carrier animals, but to a low level avoiding toxin production. Healthy carrier animals represent a reservoir of *C. botulinum*, and contribute to the dissemination of *C. botulinum* in the environment by their feces.

Group II *C. botulinum* strains are responsible for numerous botulism outbreaks in humans, and uncooked or minimally heated, chilled foods are often involved in the transmission of botulism to humans [109,110]. Albeit group II *C. botulinum* strains form spores that are less thermoresistant than those of group I *C. botulinum*, they can grow and form toxin at low temperature. Thereby, group II *C. botulinum* are a safety risk in home-made or industrial foods uncooked or processed with mild heat treatments, and notably if they are stored for long periods even at low temperature. Food contamination with group II *C. botulinum* is frequently associated with two sources from healthy carrier animals: the pork carrier of *C. botulinum* B4 and the fish carrier of *C. botulinum* E.

#### 5.2.1. *C. botulinum* B and Pork Meat

Pork meat preparations are an important risk of botulism in several countries, such as in France and Poland [111,112]. In France, type B botulism with home-made preparations of pork meat is the most prevalent foodborne botulism. Home-made or small-scale preparations of uncooked and salted ham, and to a lower extent pork meat preparations such as “pâté” with minimal heat treatment, are the most frequent sources of contamination [112,113]. The *C. botulinum* strains isolated from pork products and responsible for botulism outbreaks in humans are essentially group II *C. botulinum* B4 [114]. Pigs rarely develop any of the clinical symptoms of botulism. However, pigs can host type B *C. botulinum* in their digestive tract. Several investigations of *C. botulinum* have been performed with intestinal or fecal samples from asymptomatic pigs in different countries. The prevalence of *C. botulinum* was variable according to the countries, rearing method and season. Thereby, the prevalence in pig samples has been reported to be 24% in Germany, 80% in Japan, 62% in Sweden and 3% in Finland [115,116,117,118]. *C. botulinum* was also detected in swine feces in the US [119]. In Japan, the *C. botulinum* strains were identified as type C, and the pig carrying was associated to the presence of these bacteria in the close environment of the animals [116]. In the other countries, *C. botulinum* type B was the most prevalent. In Sweden, it was observed that the *C. botulinum* type B carrying was higher in indoors pigs than those reared outdoors, and more frequently during the winter season [117]. In contrast, in Finland a low prevalence of *C. botulinum* type B was reported in pigs reared indoors [118]. The modes of preparation and the preservation of pork meat products are important parameters in the risk of botulism transmission to humans. 

#### 5.2.2. *C. botulinum* E and Fish

The other situation concerns group II *C. botulinum*, mainly *C. botulinum* type E in the Nordic regions. Indeed, a high prevalence of *C. botulinum* type E and to a lower extent types B and F has been reported in the environment, fish and seafood of the Northern countries of the northern hemisphere (reviewed in [109]). From 1.4% to 65% of samples from fish and seafood in Nordic countries were positive for the presence of mainly *C. botulinum* type E and less frequently types B and F (reviewed in [109]). Notably, the Baltic Sea shows a high prevalence of *C. botulinum* type E (reviewed in [120,121]). Investigations in the aquatic environments of the Baltic sea indicated that 81% of sea samples and 61% of the freshwater samples contained *C. botulinum* type E [122]. The contamination by *C. botulinum* E of fish samples, mainly intestines, from the Finland coast, ranged from 10% to 40% [123]. A high prevalence of *C. botulinum* E in fish from the Baltic sea, up to 65% of positive samples, has also been reported in Denmark and Scandinavia [124,125]. *C. botulinum* E was also found in trout farms in Finland, in up to 15% of positive fish intestinal samples [126]. Albeit with a lower incidence, *C. botulinum* E is present in the sediments, fish and seafood of the other Nordic areas, such as Alaska, the USA Great Lakes, Canada, UK, Norway, Poland, Russia and Japan (reviewed in [109,120]). In the areas located at lower latitudes, *C. botulinum* types B, C, D and F are more prevalent than *C. botulinum* E in sediments, fish and sea food samples [109,120,127,128]. For example, a survey in the Atlantic coast of France showed a global incidence of *C. botulinum* of about 17% and 4% in fish and sediment samples, respectively, with a predominant contamination by *C. botulinum* type B (71% of the contaminated samples), versus *C. botulinum* type A (22.5%) and type E (9.6%) [129]. It is noteworthy that *C. botulinum* E has also been identified in the southern hemisphere [130]. The high prevalence of *C. botulinum* type E in fish and sea food notably from the Nordic countries raises the risk of botulism transmission to humans. Preparations of fish and meat without or with only mild heat treatment, preserved in anaerobic conditions at pH ≥ 5, NaCl ≤ 5%, and at temperature above 3 °C, represent a major risk of botulism by group II *C. botulinum* [110,131,132]. Experimentally, it was found that a 2 to 3 log growth of *C. botulinum* E (from about 1 to 10^2^ or 4 to 4 × 10^3^ cfu/g) in vacuum-packaged and unprocessed fish stored two weeks at 8 °C was sufficient to induce the production of BoNT/E [133]. In vacuum-packaged fresh salmon inoculated with 1 cfu/g *C. botulinum* E, toxicity was obtained after two weeks at 8 °C [134,135]. Smoked fish preserved with 3.2% NaCl and inoculated with 4 cfu/g *C. botulinum* E showed a toxicity after three weeks at 8 °C and four weeks at 4 °C [133]. 

In the natural conditions, the level of contamination is usually low, ranging from 30 to 2700 spores/kg in raw fish, and 30 to 290 spores/kg in vacuum-packed fish products [123]. The conditions and time of food storage are important parameters for the risk of human botulism. Indeed, commercial or home-made preparations of fish, sea food, or meat from marine mammals, salted, salted and air-dried, smoked, hot-smoked and vacuum packed, or fermented such as rakfisk (fermented Norwegian trout or char), have been associated with numerous outbreaks of human botulism [109,110].

### 5.3. Human Botulism and Less Frequent Types of Animal Botulism

Albeit group III *C. botulinum* are responsible for most of the animal botulism outbreaks, animals are sensitive to the other BoNT types according to the animal species (Table 1) and botulism outbreaks with the other BoNT-producing clostridia than those of group III are occasionally reported. Since the non-group III clostridia are involved in human botulism, this raises the possibility of botulism transmission from animals to humans.

Representative, uncommon botulism types in animals are listed below.

#### 5.3.1. Botulism Type E in Birds

A few outbreaks of type E botulism have been reported in farmed birds with a potential hazard for humans. Industrial farms contain several thousands of birds (10,000 to more than 20,000 per poultry flocks), and any introduction of non-identified diseased birds or apparently healthy birds carrying BoNT/E-producing clostridia in the food chain represents a real risk for humans, notably if the meat of these birds is used for preparations which can be commercialized within more than one or two weeks. Ten outbreaks of botulism type E have been identified in industrially farmed chickens in France (two in 1998, two in 1999, four in 2000, one in 2001 and one in 2002) [136]. The origin of this type of botulism was not determined. The chickens did not receive industrial food at risk, such as anything containing fish meal, and had no contact with wild birds. The decision was the destruction of the animals.

More rarely, type E botulism has been described in wild birds. Two outbreaks concerning 5000–10,000 gulls were reported in northern France in 1996. The gulls fed on fish wastes from fish industries [137]. Massive outbreaks of type E botulism in fish-eating birds have also been reported on the Great Lakes, northern America [57,138]. The mortality was estimated to 50,000 birds from 1999 to 2009 (US Geological Survey (USGS), National Wildlife Health Center (NWHC), unpublished data). Eating of moribund fish or live fish containing *C. botulinum* spores in their digestive tract seems to be the main contamination way of wild birds [138]. Maggots and water snails might also represent a reservoir of *C. botulinum* and possible transmission to waterbirds [139]. *C. botulinum* was also recovered in the sediment and plants of the Great Lakes [57].

To date, no human type E botulism associated with birds has been reported. 

#### 5.3.2. Botulism Type A and B in Cattle

Large outbreaks of type B botulism in cattle occurred in The Netherlands during the period 1976–1979 [48,140]. The origin was contaminated wet brewers’ grains that have been distributed to cattle as feeding. Proteolytic *C. botulinum* type B rapidly grows and produces toxin in brewers’ grain or grass silage [141,142]. Large numbers of *C. botulinum* were recovered in the feces of cows, where they can persist more than eight weeks. Spreading of manure on the grass pastures and preparation of grass silages from these pastures contributed to the increase of the level of contamination in the environment and the dissemination of botulism outbreaks [48]. 

A few other type B botulism outbreaks have been described in cattle fed with bale barley haylage [69], rye silage [143], plastic-packed hay [74] in the US, or maize silage in Israel [144].

Type A botulism is rare in cattle. A large outbreak of type A botulism was reported in Brazil due to the ingestion of bones and decomposing carcasses by cattle [145]. Group III *C. botulinum* was identified in 18.7% and *C. botulinum* type A in 3% of fecal samples from dairy cows, buffaloes, sheep and goats in Egypt [146].

Investigations in Sweden and Germany showed that *C. botulinum* types A and B, and to a lower extent, E and F, can be detected in the feces of healthy cattle [147,148]. The carriage of *C. botulinum* B in the Swedish cattle was higher in the winter than in the summer period [147]. 

The risk of botulism transmission to humans is based on the possible contamination of meat or milk and milk products with *C. botulinum* spores. Meat can be contaminated with fecal *C. botulinum* spores during the processing of the animals at the slaughter house. Raw milk contamination by *C. botulinum* results essentially from the cattle environment. Indeed, in farms with a botulism outbreak, the diseased as well as non-symptomatic animals excrete large numbers of *C. botulinum* spores in their feces that are subsequently spread in their local environment, including soil and pasture [48,115]. Dust represents an easy contamination way of raw milk. However, the level of contamination of milk and dairy products is usually low, for example less than 10 spores/g in mascarpone cheese [149], or less than 0.002–0.005 spores/g in dehydrated dairy ingredients [150]. Therefore, the UK Food Standards Agency recommends to not use the milk of a farm with a botulism outbreak until 14 days after the last case of botulism [151]. The excretion of BoNT in milk by diseased cows is unlikely. The toxin is very rarely detected in the serum of cattle with botulism, thus the passage of a significant amount of BoNT in the milk is improbable [152]. One report shows the presence of BoNT/B in the milk of a cow suffering from mastitis of only one quarter [153]. However, mastitis is not reported in the other observations of cows with botulism. In a survey of 51 udder milk samples from dairy cows dead with symptoms of botulism, 10 contained BoNT and seven contained *C. botulinum*, including one sample with both BoNT and *C. botulinum.* The typing of BoNT and *C. botulinum* was not determined, and it is not clear whether the infection occurred before or after death [154]. 

Milk and most of the dairy products are appropriate substrates for the growth of *C. botulinum* and toxin production [152,155]. Standard milk pasteurization is insufficient to inactivate the *C. botulinum* spores. The stability of BoNT in milk is variable according to the BoNT type and heat treatment. Conventional milk pasteurization (63 °C, 30 min) inactivates BoNT/A, whereas BoNT/B remains active [156]. However, the ultra-high temperature process (UHT) (72 °C, 15 s) almost completely inactivates both BoNT/A and BoNT/B [157]. Thus, contaminated milk and dairy products are potential sources of human botulism. A bioterrorist scenario has been considered, resulting in a large number of people poisoned with milk containing BoNT [158].

However, despite the risk associated with milk and dairy products, only a low number of outbreaks of human botulism with milk products has been reported (reviewed in [152]). Twenty outbreaks (ten of type A, seven of type B, and three undetermined) have been described since 1912. The incriminated food was home-made or commercial cheese (14 outbreaks including seven type A, five type B, and two undetermined), commercial milk (4 outbreaks including two type A, one type B and one undetermined) and yogurt (2 outbreaks including one type A and one type B) [152]. The source of contamination was largely unknown. The contamination of a commercial cheese responsible for one botulism outbreak in France was not directly due to milk contamination, but to the straw on which the cheeses were laid [159]. One infant type B botulism case was associated with *C. botulinum* B in infant formula milk powder contaminated with *C. botulinum* spores. The origin of contamination, in the factory processing or in the patient’s home, was not determined [160]. Thereby, milk contamination might occur at the farm from diseased cows or the farm environment, but also during the dairy chain process, for example by the addition of ingredients or condiments that can vehicle *C. botulinum* spores. The conditions of the storage of milk and dairy products such as temperature and duration of conservation, are key factors involved in potential bacterial growth and toxin production.

Chronic and moderate symptoms of botulism were identified in farmers who were in close contact with diseased cows. *C. botulinum* E and A were detected in 13 and 3 fecal samples, respectively, from 77 farmers, whereas *C. botulinum* A, C and D were more prevalent in feces samples from cattle, indicating a non-direct transmission from cattle to humans. However, it was shown that a diseased cow can contain more than one *C. botulinum* type. The transmission of chronic botulism to these farmers remains unclear [161].

#### 5.3.3. Botulism Type A and B in Other Animal Species

In Europe, botulism types C and D are the most prevalent forms in horses, whereas in North America, botulism type B, and to a lower extent type A, are commonly found in horses [162,163,164]. Indeed, *C. botulinum* type B is frequently identified in horses with botulism in the US [164,165,166,167,168], and only one outbreak of equine type B botulism has been reported in Europe (The Netherlands) [169]. Several outbreaks of type A botulism in horses and foals have been reported in North America [77,170,171,172,173,174]. However, no human botulism case has been associated with horse botulism.

#### 5.3.4. Botulism with Atypical BoNT-Producing Clostridium Strains

More rarely, human botulism is caused by atypical neurotoxigenic *Clostridium* strains from non-*C. botulinum* species such as *C. baratii* and *C. butyricum*. The origin of the contamination by these strains is often unknown. In some cases, a possible origin from animals is suspected. Thereby, the *C. baratii* producing type F7 BoNT is responsible for rare food borne botulism and infant botulism cases (reviewed in [175]). In three outbreaks of foodborne botulism, the incriminated food contained beef meat. One case was reported in The USA in 2001 in a woman who has consumed spaghetti with tomato meat sauce [176]. One outbreak in Spain in 2011 concerned five family members. Type F BoNT and *C. baratii* were identified in the stools of the patients. The suspected food was meat pies, but no leftovers were available for *C. baratii* investigation [177]. Another outbreak occurred in France in 2015. Three persons who have consumed a Bolognese sauce in a restaurant developed a severe botulism. Type F BoNT was detected in the serum and stool samples from the patients, and *C. baratii* was isolated from stool samples and from the ground beef meat that has been used for the Bolognese sauce preparation. The ground beef meat was from an industrial company, and 26 frozen samples from the company were negative for the presence of *C. baratii* or *C. botulinum* [178]. These observations raise the possibility that cattle, notably their digestive tracts, might be reservoirs of *C. baratii*. However, the carriage of *C. baratii* in cattle is unknown, and the meat contamination might result not directly from the intestinal content of cattle during the processing of meat, but from ingredients added in the meat preparations, such as spices.

Another human botulism risk is associated with pets and their environment. BoNT/E-producing *C. butyricum* is responsible for food borne botulism outbreaks, notably in China [4,179,180]. Neurotoxigenic *C. butyricum* strains have been isolated from fermented soybeans and soil samples around the patient home in China [180,181,182]. This pathogen is also involved in rare cases of infant botulism. Most often, the source of contamination is unknown [183,184,185,186,187]. In UK, two infant botulism cases have been associated with terrapins and/or their environment. Thereby, in one outbreak neurotoxigenic *C. butyricum* has been isolated from the fecal sample of the infant and from the tank water, tank sediment and the turtle food of the infant’s home. In the second case, the infant visited a relative who cared terrapins. The 10 days-old baby developed a botulism with neurotoxigenic *C. butyricum* in fecal sample. He was not directly exposed to terrapins or their environment. It was suggested that the transmission of *C. butyricum* spores was mediated by the person in contact with terrapins [188]. Albeit rare, these observations support that pets and their environment might be a source of human botulism, more specially in infants. 

## 6. Concluding Remarks

BoNT-producing clostridia are sporulating bacteria, mostly clostridia, which are widespread in the environment throughout the world. They synthesize potent neurotoxins which all induce a severe flaccid paralysis, but which are divided into multiple toxinotypes and subtypes. Clostridia producing certain BoNT types have a preferential geographical localization. Notably, *C. botulinum* type E is mainly distributed in northern areas of the northern hemisphere. *C. botulinum* type E belongs to group II *C. botulinum* which can grow and synthesize BoNT at low temperature. However, group II *C. botulinum* type B which shares the same physiological properties, shows a wider distribution in northern and southern areas. 

The parameters controlling the localization of these environmental bacteria remain to be better understood. BoNTs are responsible for botulism in man and vertebrate animals with the exception of a newly identified PMP1 protein targeting the Anopheles. Despite this, the widespread distribution of BoNT-producing bacteria and the abundance of botulism outbreaks in animals, contrasts with the rarity of human botulism. Man and animals show variable sensitivity to the distinct BoNT subtypes. Indeed, botulism types A, B and E, and to a lower extent F, occurs mainly in humans, whereas type C and D botulism is predominant in animals. The high sensitivity of animals to BoNT/C and/or D and the relative resistance of man to these toxinotypes likely account of the distinct epidemiology of botulism between man and animals. The high number and promiscuity of animals in each farm, as well as the access to food possibly contaminated with BoNT-producing clostridia from the environment and favorable to the development of clostridia (silages, fermented grains, etc.) amplify the incidence of animal botulism. In contrast, outbreaks of types C or D botulism in man are extremely rare, despite the high prevalence in birds and cattle farms the products of which can be introduced in the food chain for humans. No or minimally heated foods containing poultry or beef meat with a prolonged storage are potentially at risk of human botulism, whereas fresh meats have not been involved in human botulism due to the usual low level of contamination (0.1 to 7 spores/kg) [110,189]. The safety measures and controls in the food industry restrict the frequency of human botulism. Most food borne outbreaks of botulism are associated with home-made preparations [189]. Thereby, the typical types C and/or D animal botulisms are not major risks of botulism transmission to man.

However, animals can develop other botulism types or can be healthy carriers of other *C. botulinum* types than *C. botulinum* types C and D, and thus they constitute a source of human botulism. The most representative examples are healthy carrier animals of group II *C. botulinum* including *C. botulinum* E in fish and *C. botulinum* B4 in pigs. Albeit animals are sensitive to BoNTs from group I *C. botulinum*, they only occasionally develop these botulism types. It is noteworthy that animals are rarely involved in transmission to humans of botulism from group I *C. botulinum*. Moreover, only very rare observations suggest that animals or their environment can transmit botulism to humans with atypical BoNT-producing clostridia such as *C. butyricum* E and *C. baratii* F. BoNT-producing clostridia are essentially environmental bacteria adapted to survive very long periods. The evolution of these bacteria in multiple BoNT types and subtypes remains mysterious. It is questionable whether this evolution in variable BoNT types represent an initiator step in adaptation to specific animal and human hosts with limited interspecies transmission (Figure 1). The major risk of transmission of botulism from animals to humans is by healthy carrier animals of certain *C. botulinum* types, whereas the apparently adapted *C. botulinum* C and D to animals are rarely observed in man.

## Figures and Tables

**Figure 1 toxins-12-00017-f001:**
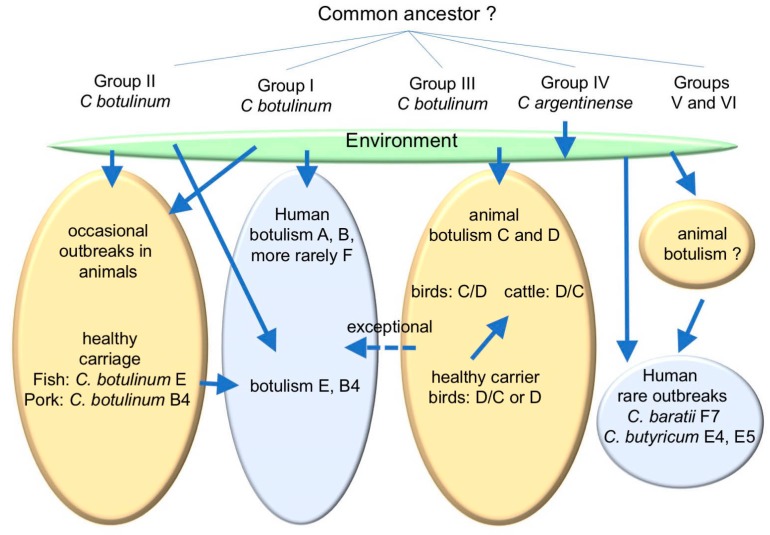
Schematic representation of the evolution of BoNT-producing clostridia and main possible transmission pathways between environment, animals, and humans. Note that there is no direct transmission of botulism from diseased or healthy carrier animals to humans. Food borne botulism occurs by food contamination combined with food preservation under conditions favorable for toxin production. Infant botulism might result from ingestion of *C. botulinum* spores directly from the environment.

**Table 1 toxins-12-00017-t001:** Toxicity of botulinum toxin types according to the animal species and injected intraperitoneally, or as indicated. The toxicity titers are expressed as intraperitoneal mouse lethal doses (MLDs) per kg of body weight, or as indicated, according to [40] and from indicated references.

BoNT Type (Human, Animal Species)	A	B	C	C/D	D	E	F
Human	~1 ng/kg (parenteral) [41,42] ~1 μg/kg (oral) [41,42] 5000–12,000/adult (oral) [43]	3000–4000/adult (oral) [43]					
Monkey	400 650 (oral) [44] 30,000 (oral) [45] 39 (IM) [46] 40 (IV) [45]	180 (oral) [44]	>9000 >100,000 (oral) [44]		>70,000 >600,000 (oral) [44] 10^6^ (oral) [44]	150–200 (SC) [44] 1500–2500 (oral) [44] 8000 (oral)	25–50 (SC) [44] 50,000–75,000 (oral) [44] 150,000 (oral)
Horse	1800 (SC)		>8000 (SC)		20,000 (oral)		
Mice	1	1	1		1	1	1
Mice 1 LD50	7.0 pg [47]	20.2 pg [47]				24.5 pg [47]	48.2 pg [47]
Rat	10–25						
Guinea pig	5	4	1.6–30		4	34	25–50 (SC) [44]
Rabbit	25–40		20		12 (SC)		
Cattle		5.6 × 10^7^ (oral) [48]	4 [49]		2.2 (SC) 22.5 (oral)		
Swine	20,000 (IV)	180 (IV) 3 × 10^6^ (oral)	>18,000 (IV) >0.3 × 10^6^ (oral)		>67,000 (IV) >0.78 × 10^6^ (oral)	14,000 (IV) 1.4 × 10^6^ (oral)	4000 (IV) 0.17 × 10^6^ (oral)
Cat	12,000		40,000 (SC)		7.5 × 10^7^ (SC)		
Dog	18,000		1000 [50]		100,000 [50]	100 [50]	
Mink	1000–24,000 10^7^ (oral)	100,000 >10^6^ (oral)	1000 40–167 (SC) 10^4^–10^5^ (oral)		10^7^ >10^8^ (oral)	10,000 10^7^ (oral)	10^6^ >10^6^ (oral)
Ferret			10^6^ (oral)				
Chicken	10 (IV) 10^6^/kg (oral) [49]	20,000 (IV)	16,000 (IV) 2.6 × 10^4^ (IV)/animal [11]	5.6 × 10^3^ (IV)/animal [11]	>320,000 (IV) 8 × 10^7^ (IV)/animal [11]	100 (IV)	640,000 (IV)
Duck		1.5 × 10^4^ (IV)/duck [11] >10^7^/duck (oral) [51]	500–76,000 45–80,000 (Oral) 19,000–320,000 (oral per bird) 9.6 × 10^4^ (ip)/duck [11] 320,000/kg (oral) [11]			2.5 × 10^6^ (IV)/duck [11] > 2 × 10^5^/duck (oral) [51]	
Peafowl	170 (IV)	33,000 (IV)	2700 (IV)		>320,000 (IV)	170 (IV)	640,000 (IV)
Pheasant	44–170 (IV) 440,000/kg (oral) [51]	88,000 (IV)	70 (IV)		>320,000 (IV)	440 (IV) 440,000/kg (oral) [51]	640,000 (IV)
Turkey	20 (IV) 200,000/kg (oral) [51]	40,000 (IV)	320 (IV)		>320,000 (IV)	200 (IV) 200,000/kg (oral) [51]	640,000 (IV)

SC, subcutaneous; IM, intramuscular; IV, intravenous.

**Table 2 toxins-12-00017-t002:** Typing of animal botulism from samples received in the laboratory Anaerobic Bacteria and Toxins during the period 1998–2012 in France by toxin identification with mouse bioassay and specific neutralizing anti sera. Neutralization by both anti-BoNT/C and D likely corresponds to genotype C/D, and neutralization by anti-BoNT/D to genotype D or D/C [11,12].

Animal Species	Positive Samples	Toxin Neutralization				
		Anti-Type C	Anti-Type D	Anti-Types C and D	Anti-Type E	Undetermined
Farmed birds ^a^	517	87	129	236	10	55
Wild birds ^b^	385	170	72	126	0	17
Bovine ^b^	125	16	28	24	0	5

^a^ Serum samples from farmed birds (40% chickens, 32% turkeys, 14% ducks, 14% other); ^b^ Toxin identification in supernatants of enrichment cultures from intestinal content [65]. Wild birds were mostly ducks (87%).

**Table 3 toxins-12-00017-t003:** Human Botulism Type C and D.

Human Botulism Type C						
Country	Year	Outbreaks	Cases	Origin	Clinical Aspect	Reference
USA	1950	1	1	possibly food	BoNT/C and *C. botulinum* C in the stomach of a person died with symptoms of botulism from a family group of four persons	[97,105]
France	1955	1	2	home-made “pâté” isolation of a *C. botulinum* type C strain	Two patients with a moderate form of botulism from a group of eight persons	[106]
Rhodesia	1960	1	4	home-made “pâté” isolation of a *C. botulinum* type C or B strain	Four patients with a moderate form of botulism	[105,107]
USRR	1965, 1966	2	2	no reported	no reported	[108]
France	1972	1	4	possibly smoked chicken	Two severe forms, including one decease by infectious complications, BoNT/C in the serum samples of the two patients, and two mild forms	[99]
Japan	1990	1	1	possibly environmental contamination	infant botulism (171 days-old female) BoNT/C and isolation of a *C. botulinum* type C strain from stool sample	[96]
Guyana	2006	1	1	diseased chicken	One mild form with symptoms of botulism	[100]
**Human Botulism Type D**						
Chad	1958	1	6	home-made ham BoNT/D and isolation of a *C. botulinum* type D strain	Six mild forms from a group of eight. Five patients with food poisoning symptoms (vomiting, diarrhea), one developed blurred vision. One patient with typical symptoms of botulism (diplopia, dry mouth, dysphagia, constipation, asthenia) recovery in one month	[101,102]

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
