# Peer review of "Public Health Risk Associated with Botulism as Foodborne Zoonoses"

_toxins, 2019, doi:10.3390/toxins12010017_

Round 1
Reviewer 1 Report
This is a very comprehensive review article. It would benefit greatly from editing to correct the English language.
Questions/comments:
The authors should explain how animal botulism occurs. Does it occur as a foodborne intoxication (eg. cattle consuming a hay bale with a decaying cat), or does it occur as colonization of the gut by C. botulinum with concurrent production of botulinum toxin? Does it occur in the same manner in different species? Does C. botulinum multiply in the animal intestinal tract, allowing animals to act as reservoirs for the organism by shedding the organism in feces? Spores of C. botulinum are widespread in the environment and persist for long periods of time. The finding of spores of C. botulinum in intestinal contents of animals (eg C. botulinum type B in pigs; C. botulinum type E in or on fish) does not necessarily constitute a risk to humans. The spores need to contaminate flesh, grow, and produce botulinum toxin in order to cause human botulism. This is an important difference compared to several other foodborne pathogens where ingestion of the pathogen causes gastrointestinal illness in the human host (eg. VTEC, Salmonella, Campylobacter, Listeria). The author(s) should consider the use of tables to summarize the incidence and levels of C. botulinum in different animals. I emphasize both incidence and levels, as isolation of C. botulinum without enumeration does not contribute very much information to formulate the risk to human health.Author Response
This is a very comprehensive review article. It would benefit greatly from editing to correct the English language.
Questions/comments:
The authors should explain how animal botulism occurs. Does it occur as a foodborne intoxication (eg. cattle consuming a hay bale with a decaying cat), or does it occur as colonization of the gut by C. botulinum with concurrent production of botulinum toxin? Does it occur in the same manner in different species? Does C. botulinum multiply in the animal intestinal tract, allowing animals to act as reservoirs for the organism by shedding the organism in feces?
Animal botulism can result from ingestion of preformed toxin in food or from intestinal colonization and toxin production in the intestine. Contamination of food by a carrion such as small rodents or cats, that is an appropriate environment for C. botulinum growth and toxinogenesis, and decaying carcasses that are chewed by phosphorus-deficient cattle are the main source of botulism intoxication in animals. Toxin production can be found in fermented cereals or silage with pH > 4.5, but in most outbreaks of bovine botulism, botulinum toxin was no detected in silages or industrial livestock feed mainly based on dry crushed grains containing C. botulinum spores that were responsible for botulism. Botulism by intestinal colonization is a frequent form of botulism in farmed birds and cattle. Indeed, botulism cases are observed up to two weeks after the withdrawal of the contaminated food.
It is difficult to said whether the different animal species show different susceptibility to intestinal colonization by C. botulinum. No data are available on this aspect. The age of animals seems also a susceptibility factor for intestinal colonization as in humans. Botulism by intestinal colonization is more prevalent in foals whereas food borne botulism seems more frequent in adult horses.
These comments have been added in the revised manuscript lines 176-195.
In healthy carrier animals, C. botulinum can possibly grow in the intestinal content, but to a low level avoiding toxin production. This contributes to the dissemination of C. botulinum in the environment of these animals by their feces. A comment has been added lines 281-285. This is the case of chickens that are resistant to botulism type D and can cause botulism outbreaks in cattle by spreading the litter on pastures. No data are available on the growth of C. botulinum in the digestive tract of healthy carriers.
Spores of C. botulinum are widespread in the environment and persist for long periods of time. The finding of spores of C. botulinum in intestinal contents of animals (eg C. botulinum type B in pigs; C. botulinum type E in or on fish) does not necessarily constitute a risk to humans. The spores need to contaminate flesh, grow, and produce botulinum toxin in order to cause human botulism. This is an important difference compared to several other foodborne pathogens where ingestion of the pathogen causes gastrointestinal illness in the human host (eg. VTEC, Salmonella, Campylobacter, Listeria).
We agree with this comment that C. botulinum spores in intestinal contents of animals do not represent a direct risk of botulism to humans. Contamination of food prepared from animals and conditions of storage of foods allowing toxin production are key factors for botulism risk in humans. This has been highlighted in the manuscript, and the following sentences have been added in the legend of Fig. 1: "Note that there is no direct transmission of botulism from animal healthy carriers to humans. Transmission to humans occurs via food contamination and food preservation in conditions favorable for toxin production".
The author(s) should consider the use of tables to summarize the incidence and levels of C. botulinum in different animals. I emphasize both incidence and levels, as isolation of C. botulinum without enumeration does not contribute very much information to formulate the risk to human health.
We agree that the incidence of botulism in animals and the levels of C. botulinum rather than the only detection of this pathogen are important information to estimate the risk to human health. Unfortunately, there are very few reports about quantitative incidence of animal botulism and enumeration of C. botulinum in animals. Survey of animal botulism is highly variable in the different countries. There are only estimations of the number of cases in a few botulism outbreaks in animals. Some examples of animal botulism outbreaks have been added lines 162-170 and 244-249.
Reviewer 2 Report
The manuscript was well organized and assembled some obscure and hard to find references concerning the rare cases of Group III intoxication in humans. The authors also did a very comprehensive job of summarizing any potential zoonotic transmission examples of BoNT intoxication. In my mind there were few points of substantial contention. Lines 120-122 state "Thereby, all human and vertebrates are susceptible to develop botulism A-G". The mosquito specific variant does not mean all vertebrates are susceptible and it reads as if they are subject to A-G as well. The spacing on the description of Table 1 should be reviewed, spaces need to be inserted.
I do feel the manuscript would benefit from an English language review. Otherwise, I thought it was a good, solid manuscript.
Author Response
The manuscript was well organized and assembled some obscure and hard to find references concerning the rare cases of Group III intoxication in humans. The authors also did a very comprehensive job of summarizing any potential zoonotic transmission examples of BoNT intoxication. In my mind there were few points of substantial contention.
Lines 120-122 state "Thereby, all human and vertebrates are susceptible to develop botulism A-G". The mosquito specific variant does not mean all vertebrates are susceptible and it reads as if they are subject to A-G as well.
The paragraph has been modified as follows:
"BoNT/A to G can attack all the vertebrates, but the invertebrates are resistant due to lack of specific receptors on their neuronal cell surface [35]. Thereby, human and all vertebrates are susceptible to develop botulism A to G, but the sensitivity of each animal species is variable according to the BoNT type (Table 1). The host sensitivity depends of the presence/abundance of functional BoNT receptors on neuronal cell surfaces and also of the presence or absence of cleavable intracellular target isoforms (SNAP25, VAMP). The sensitivity of the main domestic animal species to BoNTs according to experimental tests is summarized in Table 1. Recently, a novel BoNT called paraclostridial mosquitocidal protein1 (PMP1) produced by a Paraclostridium bifermentans subsp Malaysia strain has been found to be specific of Anopheles mosquitos [36]. This is the first BoNT type that has been identified to target an invertebrate species."
The spacing on the description of Table 1 should be reviewed, spaces need to be inserted.
Table 1 format has been revised by Toxins and will be further checked for the final version
I do feel the manuscript would benefit from an English language review. Otherwise, I thought it was a good, solid manuscript.
The manuscript will be further checked for English language by the editors of Toxins.